

# Acoustic individuality in the hissing calls of the male black grouse (*Lyrurus tetrix*)

Lucie Hambálková, Richard Policht, Jiří Horák and Vlastimil Hart

Department of Game Management and Wildlife Biology, Faculty of Forestry and Wood Sciences, Czech University of Life Sciences, Prague, Czech Republic

## ABSTRACT

Acoustic individuality may well play a big role during the mating season of many birds. Black grouse (*Lyrurus tetrix*) produces two different long-distance calls during mating on leks: rookooing and hissing calls. The first one represents low frequency series of bubbling sounds and the second one represents hissing sound. This hissing represents a signal not produced by the syrinx. We analyzed 426 hissing calls from 24 individuals in Finland and Scotland. We conducted cross-validated discrimination analyses (DFA). The discrimination model classified each call with almost 78% accuracy (conventional result) and the validated DFA revealed 71% output, that is much higher than classification by chance (4%). The most important variables were Frequency 95%, 1st Quartile Frequency, Aggregate Entropy and Duration 90%. We also tested whether between individual variation is higher than within individual variation using PIC (Potential for individual coding) and we found that all acoustic parameters had PIC > 1. We confirmed that hissing call of black grouse is individually distinct. In comparison to the signals produced by the syrinx, non-vocal sounds have been studied rarely and according to our knowledge, this is the second evidence of vocal individuality in avian hissing sounds which are not produced by syrinx. Individuality in the vocalization of the male black grouse may aid females in mating partner selection, and for males it may enable competitor recognition and assessment. Individually distinct hissing calls could be of possible use to monitor individuals on leks. Such a method could overcome problems during traditional monitoring methods of this species, when one individual can be counted multiple times, because catching and traditional marking is problematic in this species.

## INTRODUCTION

At the time of a decline in the black grouse (*Lyrurus tetrix*) population across its distribution range (*Ciach, 2015*; *Jankovska et al., 2012*; *Kasprzykowski, 2002*), determining actual number of individuals is becoming increasingly important. The causes for this decline vary: change of the environment and climate (*Kurhinen et al., 2009*; *Kvasnes et al., 2010*; *Viterbi et al., 2015*; *White, Warren & Baines, 2013*), parasite infestation (*Jankovska et al., 2012*), predation (*Charnov, Orians & Hyatt, 1976*; *Korpimaki, Koivunen & Hakkarainen, 1996*; *Pekkola et al., 2014*; *Policht et al., 2019*; *Summers et al., 2004*; *Tornberg et al., 2013*; *Widen et al., 1987*), reducing genetic diversity (*Segelbacher, Hoglund &*

Corresponding author
Lucie Hambálková,
hambalkoval@fld.czu.cz

*Storch, 2003*; *Segelbacher et al., 2014*) and human activities (*Formenti et al., 2012*; *Hess & Beck, 2012*; *Ingold, 2005*; *Nichter, Lipp & Gregory, 2017*; *Storch, 2013*). Methods and options for protection and conservation of grouse are manifold and are realized at the local, regional, and national levels (*Storch, 2013*).

An integral part of any conservation measures in general, and thus also in the case of black grouse populations in particular, is monitoring. Methods of monitoring are diverse. *Franceschi et al. (2014)* simulated two monitoring approaches: plot sampling and distance sampling. According to their study, distance sampling is a better way to monitor grouse in terms of accuracy. On the other hand, this approach is also costly, as it requires 4–5 sampling points per km$^2$ for reliable outcomes (*Franceschi et al., 2014*). The most common counting method for black grouse is to register males displaying in the spring (*Hancock et al., 1999*). Depending on the size of the area to be monitored, it is possible to perform a full-area survey or to select sample areas at random (*Hancock et al., 1999*). Monitoring based on acoustic features of bird vocalization could be a more efficient method. *Laiolo et al. (2007)* recommended combining counting based on vocalization with physical marking.

Black grouse males produce the following kinds of sounds: resonant rookooing call and hissing calls. The latter is further subdivided into tones of aggression and alarm calls (*Cramp, 1983*). This study is focused on a particular type of hissing call—crowing-hiss, described by *Cramp (1983)* as harsh and angry sound, which is produced during the display of male black grouse. This hissing sound is not produced by syrinx. Such non-vocal sounds are produced by some constriction located on the way from the lungs to the bill (*Fitch & Hauser, 2003*). Potential information encoded in non-vocal sounds of birds remains almost unstudied (*Budka et al., 2018*). Recent research of hissing sounds produced by geese confirmed encoding of individual identity during antipredator behavior (*Policht et al., 2020*). In comparison to non-vocal acoustic signals of birds, majority of bioacoustic studies focused on research of sounds produced by syrinx. A hissing sound also appears in black grouse chicks above the age of 3 weeks, but we do not suppose it is the same sound category that is the focus of our study (*Meinert & Bergmann, 1983*). This type of vocalization, along with the rookooing call, is the most prominent sound made by black grouse, which can be heard over long distances and even in closed habitats such as forests with dense undergrowth. Such calls are frequently used for population monitoring to find actually used leks and counting present males. Therefore, this type of call may play an important role in noninvasive monitoring of black grouse. The rookooing call can be characterized as a low-frequency, repetitive sound within a range of about 200 to 1,000 Hz. This is why this type of call often overlaps with background noise frequencies. Compared with this, the hissing call is found in the frequency range of 350 to 4,500 Hz and is therefore easier to filter out from background noise and to mark this type of call for measurement using acoustic software. Thanks to these characteristics, the hissing call may be more suitable for acoustic monitoring of black grouse.

In an effort to ensure quiet conditions for game wildlife, non-invasive monitoring, such as that based on vocalization, is the method of choice. This method relies on distinguishing individuals without physical marking. In our study, we analyzed the

vocalization of male black grouse to examine variation between individuals, and to find out whether vocalization characteristics could serve as a unique identifying trait.

## METHODS

### Study areas and recording

We recorded the hissing calls of male black grouse during their mating season. Recording took place in Finland in 2012 and 2013, and in Cairngorms National Park, Scotland in 2019. Field experiments were approved by the Department of Natural Resources, Ministry of Agriculture and Forestry, Finland and by the Game & Wildlife Conservation Trust, Scotland. According to Finnish legislation in general and to the hunting legislation, this type of scientific project does not require any special permits or licenses. All appropriate permissions were in place for the fieldwork in Scotland. The research was conducted in accordance with the guidelines of the Animal Behavior Society for the ethical use of animals in research. The study was carried out in accordance with the recommendations in the Guide for Care and Use of Animals of the Czech University of Life Sciences, Prague. The Animal Care and Use Committee of the Czech Ministry of the Environment approved the protocol (Permit number: 15106/ENV/14-825/630/14).

Vocalization of male black grouse was recorded with the audio recorder Olympus LP-100 in combination with a Sennheiser ME 66 directional microphone (frequency response 20 Hz–20 kHz ± 2.5 dB) complemented by a K6 powering module. Recordings were saved in .wav format (48 kHz sampling rate, 16-bit sample size). We recorded all individuals in the wild during courtship at leks. Lek is an area where two or more males perform courtship displays to gain an advantage for mating with females. All leks were approached before the arrival of males, about 2 h before sunrise. Each recording session took on average 1 h and was performed from a portable hide so that the males could be observed without being disturbed. The distance of the hide from display sites was 10 m on average. During the pilot study, we only tested the variability between multiple individuals on one lek, and it turned out that the individual variability is much larger. To avoid the risk of multiple counting of the same individual, we chose the option of selecting only one, maximum of two individuals on each lek. The distance between visited display sites was at least one km and, according to *Borecha, Willebrand & Nielsen (2017)*, black grouse males show strong fidelity to their lek; therefore, the risk of recording the same individual at the two display sites was low.

### Acoustic analyses

Recordings were analyzed using Raven Pro 1.5 software with a 512 sample size and a Hann window. We selected good quality calls with high signal to noise ratio, non-overlapping with other hissing calls or background noise and wind. Each selected hissing call was manually bounded by the selection frame that is defined by the beginning and end of the signal and the lowest and highest frequency of the signal. Temporal and frequency variables were then measured automatically. These measurements were entered into the statistical analysis.

## Statistical analyses

We analyzed 426 good-quality calls from 31 individuals (at least ten separate hissing calls per individual). We measured 29 variables (Table 1). We excluded variables with low or no variation. The remaining variables were standardized using Z-score transformation (subtracting the mean and dividing by standard deviation). In order to test individual variation, we used stepwise Discrimination Function Analysis (DFA) using IBM SPSS Statistics 24.0 software (IBM Corp., Armonk, NY, USA). We applied a leave-one-out cross-validation procedure (IBM SPSS Statistics 20) to validate the results of DFA.

To test the potential for individual variation (Potential of Individual Coding—PIC) for each parameter, we compared the coefficient of variation (CV) within and between individuals. The PIC ratio was computed for each acoustic parameter by dividing the $CV_{between}$ by the mean of the $CV_{intra}$ values related to each individual (*Robisson, 1992*). For these tested parameters, a PIC value greater than one means that an inter-individual variability is higher than intraindividual variability. We tested a significance using Kruskal–Wallis test.

## RESULTS

### Hissing call description

The hissing calls of black grouse represent wideband acoustic signals, in which energy is spread across a wide frequency range. The duration of such calls ranged from 0.1 to 1.21 s (0.76 ± 0.16, mean ± SD). This type of call can consist of one or two notes; however, the occurrence of a two-syllable form was rare (~$n < 1$%)—so we did not analyze these calls.

The Low frequency ranged from 352.9 to 1,310.3 Hz (830.2 ± 195.6 Hz, mean ± SD) and the High frequency from 1,702.4 to 4,482.8 Hz (2,687.5 ± 536.4 Hz, mean ± SD) for all individuals. Frequency range reached 729.6 to 3,241.4 Hz (1,857.3 ± 478.6 Hz, mean ± SD). The spectrograms of black grouse recorded in Finland and Scotland are shown in the figures below (Figs. 1 and 2). The spectrograms were generated in Avisoft-SASLab Pro with FFT length, 1,024 sample size, a Hamming window and 87.5% overlap. For a representative recording of a hissing call of one individual form Scotland and one individual from Finland see Audio S1 and Audio S2.

### Individual variation

From selected parameters the resulting model (see Table S1) included 13 significant acoustic variables ($p < 0.001$; r ≤ 0.87): 1st Quartile Frequency, Relative 1st Quartile Frequency, Aggregate Entropy, Average Entropy, Relative Center Time, Call Duration, Duration 90%, Frequency 5%, Relative 3rd Quartile Frequency, Frequency 95%, Inter-Quartile Range Bandwidth, Inter-Quartile Range Duration and Time 5% (Table 1). The first four discriminant functions had Eigenvalues > 1 and explained 79.7% of the variation. With the first discrimination function mostly correlated F95% (Frequency 95%) (r = 0.767) and Q1F (Quartile 1 Frequency) (r = 0.707) and the second discriminant function correlated best with AggEnt (Aggregate Entropy) (r = 0.390) and Dur 90% (Duration 90%) (r = 0.387) (Fig. 3). The Discriminant Function Analysis excluded seven

**Table 1 Descriptions of acoustic parameters measured in Raven Pro 1.5 that entered statistical analysis.**

| Acoustic parameter name | Abbreviations (Units) | Description |
|---|---|---|
| *1st Quartile frequency | Q1 Freq (Hz) | The frequency that divides the signal into two frequency intervals containing 25% and 75% of the energy in the signal. |
| *Relative 1st quartile frequency | Q1 Freq rel, | The frequency that divides the signal into two frequency intervals containing 25% and 75% of the energy in the signal relative to the frequency range of the signal. |
| *3rd Quartile frequency | Q3 Freq (Hz) | The frequency that divides the signal into two frequency intervals containing 75% and 25% of the energy in the signal. |
| Relative 3rd quartile frequency | Q3 Freq rel, | The frequency that divides the signal into two frequency intervals containing 75% and 25% of the energy in the signal relative to the frequency range of the signal. |
| 1st Quartile time | Q1 Time (s) | The time that divides the signal into two time intervals containing 25% and 75% of the energy in the signal. |
| Relative 1st quartile time | Q1 Time rel, | The time that divides the signal into two time intervals containing 25% and 75% of the energy in the signal relative to signal duration. |
| Relative 3rd quartile time | Q3 Time rel, | The time that divides the signal into two time intervals containing 75% and 25% of the energy in the signal relative to signal duration. |
| *Aggregate entropy | Agg entropy (bits) | The aggregate entropy measures the disorder in a sound by analysing the energy distribution. Higher entropy values correspond to greater disorder in the sound whereas a pure tone with energy only one frequency bin would have zero entropy. It corresponds to the overall disorder in the sound. |
| *Average entropy | Avg Entropy (bits) | This entropy is calculated by finding the entropy for each frame in the signal and then taking the average of these values. |
| Bandwidth 90% | BW 90% (Hz) | The difference between the 5% and 95% frequencies. |
| Center frequency | Center freq (Hz) | The frequency that divides the signal into two frequency intervals of equal energy. |
| Center time | Center time (s) | The point in time at which the signal is divided into two time intervals of equal energy. |
| *Relative center time | Center time rel, | The point in time at which the signal is divided into two time intervals of equal energy relative to the signal duration. |
| *Call duration | Duration (s) | The difference between begin time and end time for the signal. |
| *Duration 90% | Dur 90% (s) | The difference between the 5% and 95% times. |
| *Frequency 5% | Freq 5% (Hz) | The frequency that divides the signal into two frequency intervals containing 5% and 95%. |
| Relative frequency 5% | Freq 5% rel, | The frequency that divides the signal into two frequency intervals containing 5% and 95% relative to frequency range. |
| *Frequency 95% | Freq 95% (Hz) | The frequency that divides the signal into two frequency intervals containing 95% and 5%. |
| Relative Frequency 95% | Freq 95% rel, | The frequency that divides the signal into two frequency intervals containing 95% and 5% relative to frequency range. |
| *Inter-quartile range bandwidth | IQR BW (Hz) | The difference between the 1st and 3rd quartile frequencies. |
| *IQR (Inter-quartile range) duration | IQR Dur (s) | The difference between the 1st and 3rd quartile times. |
| Max entropy | Max entropy (bits) | Maximum entropy calculated from each frame. |
| Max frequency | Max freq (Hz) | The frequency at which max power occurs within the signal. |
| Max time | Max time (s) | The first time in the signal at which a spectrogram point with power equal to max power/peak power occurs. |
| Min entropy | Min entropy (bits) | The minimum entropy calculated for a spectrogram slice within the signal bounds. |
| Peak time | Peak time (s) | The first time in the signal at which a sample with amplitude equal to peak amplitude occurs. |

(Continued)

| Table 1 (continued) | | |
|---|---|---|
| **Acoustic parameter name** | **Abbreviations (Units)** | **Description** |
| *Time 5% | Time 5% (s) | The time that divides the signal into two time intervals containing 5% and 95%. |
| Relative time 5% | Time 5% Rel, | The time that divides the signal into two time intervals containing 5% and 95% relative to signal duration. |
| Relative time 95% | Time 95% Rel, | The time that divides the signal into two time intervals containing 95% and 5% relative to signal duration. |

**Note:**
13 parameters (*) were included in resulting DFA model.

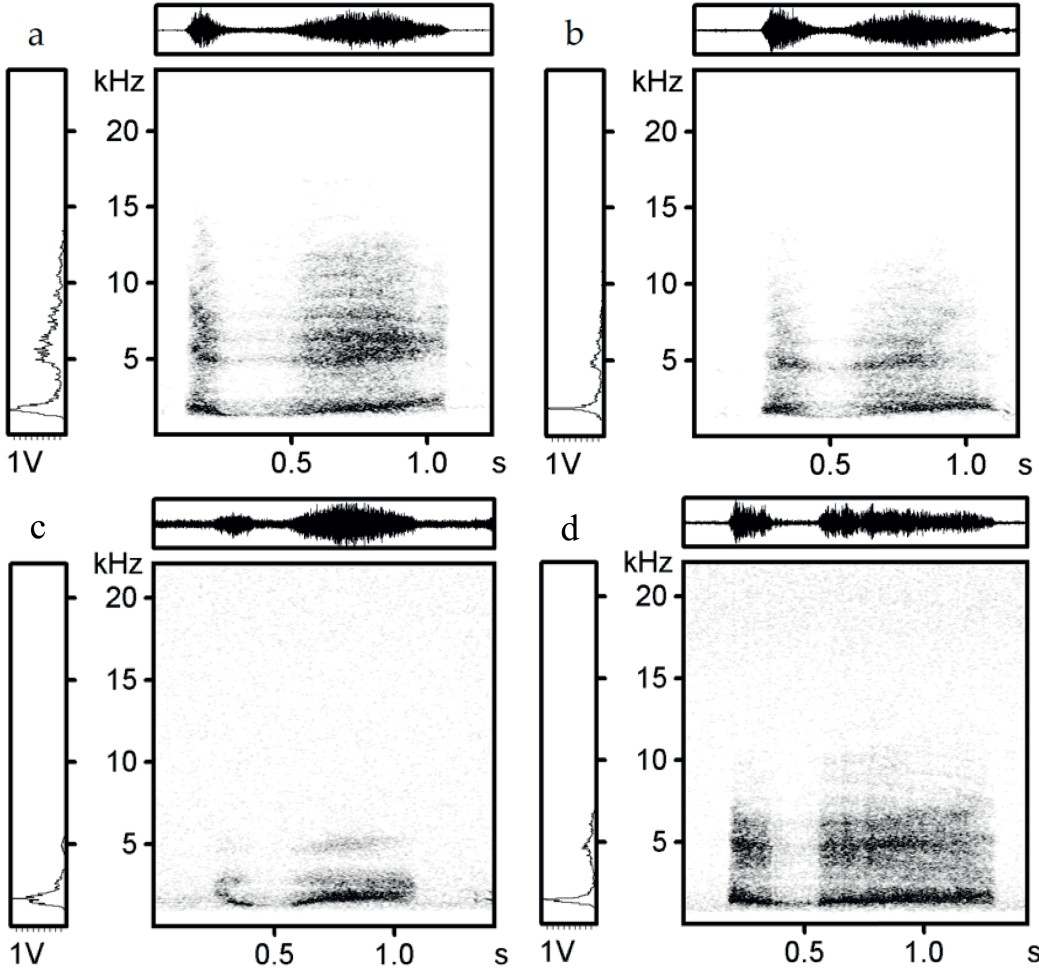

**Figure 1 Spectrograms and oscilograms: Representative hissing calls of two of the black grouse from Scotland (A, B) and Finland (C, D).** Each lettered panel refers to one individual bird. Spectrograms indicate observable differences between four individuals.

out of 31 individuals due to their missing or extreme values of the measured parameters. The cause could be a poorer degree of sound quality that did not pass the analysis. This selection has been made by model procedure automatically. The resulting DFA model correctly classified 77.9% (71.1%, cross-validated result) hissing calls. Six individuals showed the highest classification accuracy (80–100%), and most individuals ($N = 15$) were

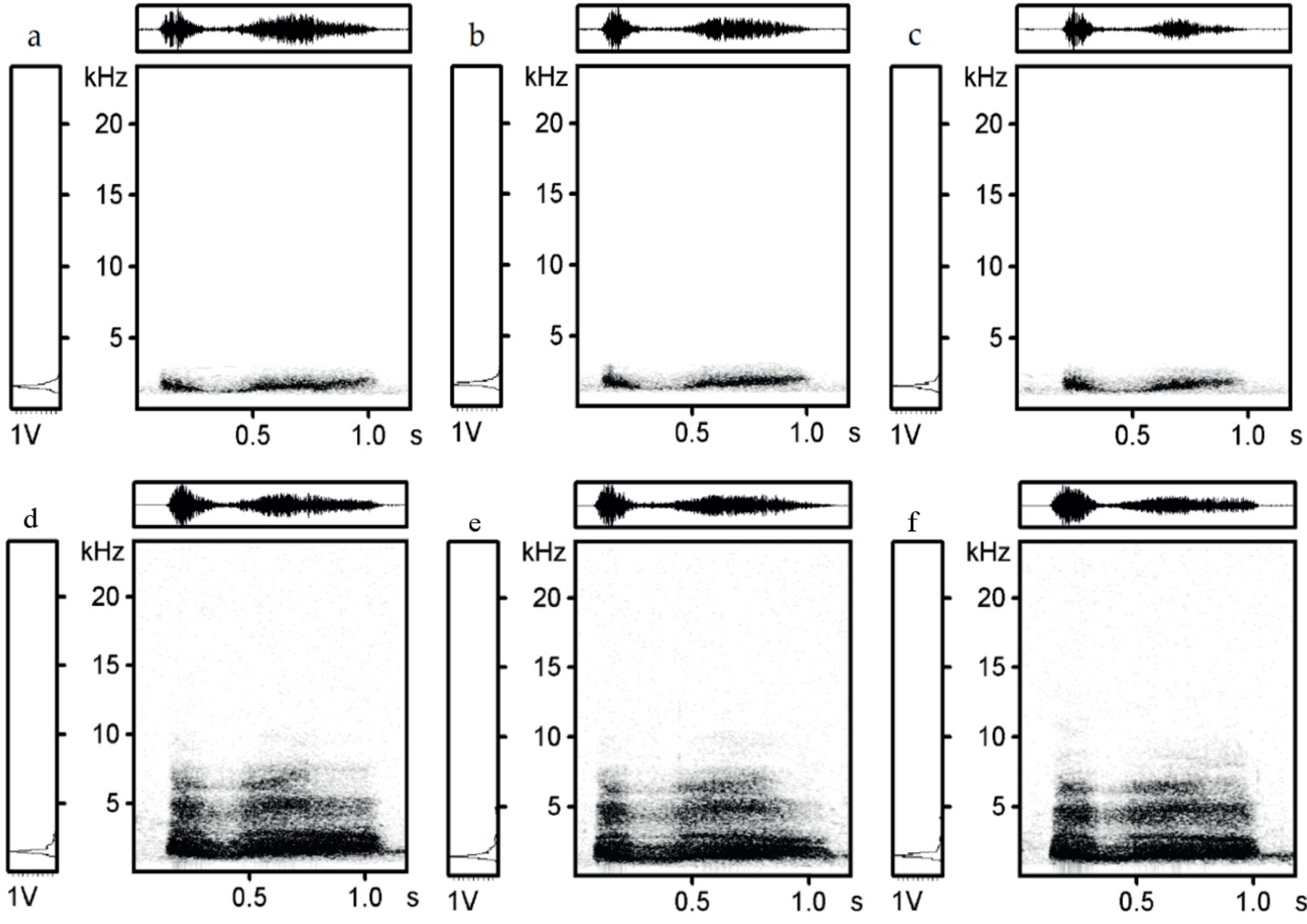

**Figure 2** **Spectrograms and oscilograms of three hissing calls of black grouse from one individual from Scotland (A–C) and one individual from Finland (D–F).** Each lettered panel refers to one hissing call. Spectrograms in rows indicate consistent stability of individual pattern within the same individual.

classified with 60–79% success. Only three males were classified with a lower than 59% success. These results were much higher than classification by chance (4%). The output of classification results is shown in Table S2. We tested whether observed classifications differed from the expected classifications (by chance) and we found a significant difference: Chi-Square = 307.1, df = 23, $p < 0.001$. We also tested whether between individual variation is higher than within individual variation using PIC and we found that all acoustic parameters had PIC > 1 (Table 2).

## DISCUSSION

Our results reveal that the wideband hissing call of male black grouse is individual specific. The discrimination model classified each call with almost 78% accuracy, and the first four discriminant functions explained nearly 80% of variation. The PIC ratio was higher than one for all parameters tested, demonstrating that the variability between individuals was higher than the variability within individuals. Therefore, the hissing call is

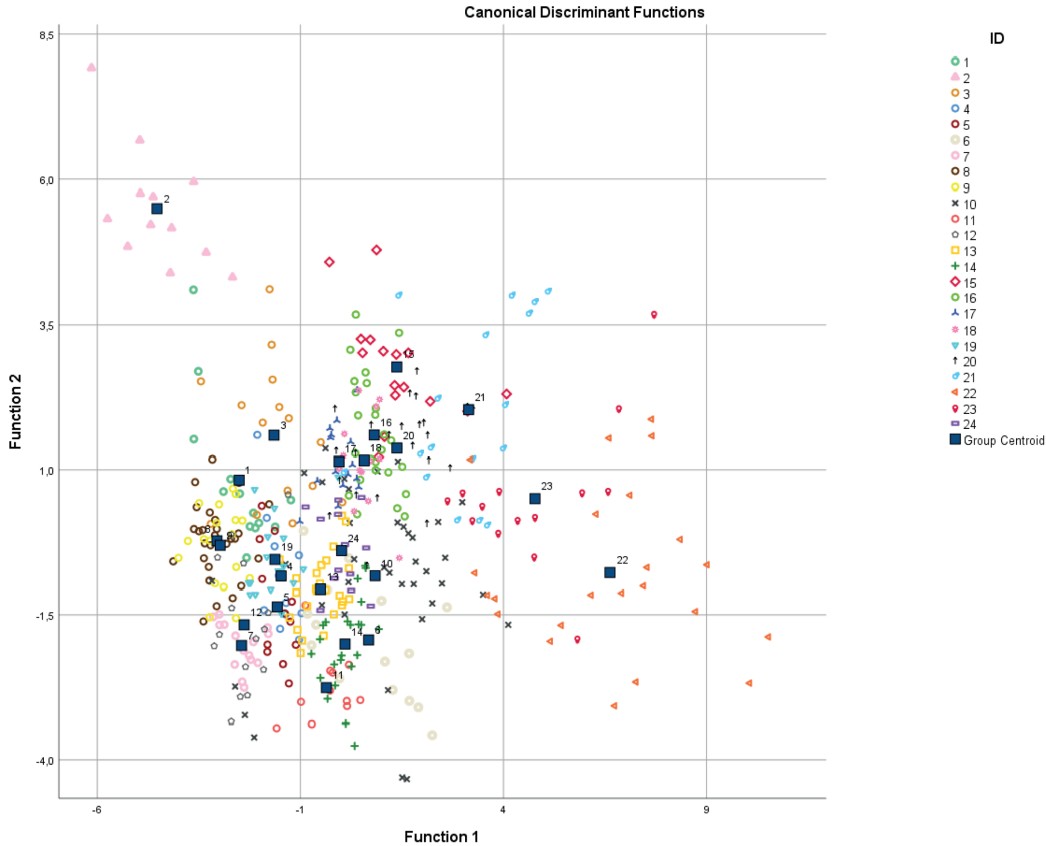

**Figure 3 Scatter plot of hissing calls.** Numbers refer to individuals, squares represent group centroids. Discrimination function 1 represents Frequency 95% and function 2 represents 1st Quartile Frequency.

a type of vocalization that carries information about individuality. Considering this type of call is a wideband, non-vocal sound, such a result is quite unique. There are not many studies focused on non-vocal animal sounds and even fewer of them have provided results confirming individuality in this type of vocalization; however, there are several. Individual variation was found in the male Houbara Bustard (*Chlamydotis undulata undulata*), which produces sounds called booms during courtship (*Cornec, Hingrat & Rybak, 2014*). Acoustic variation between individuals was also investigated in the Greater Prairie-chicken (*Tympanuchus cupido*) and the boom vocalization of this species was found to have individual characteristics (*Hale, Nelson & Augustine, 2014*). Thanks to temporal patterns, along with the number of drumming strokes, it is possible to discriminate individuals of the Great Spotted Woodpecker (*Dendrocopos major*) (*Budka et al., 2018*). According to acoustic analysis in the male Greater Sage-grouse (*Centrocercus urophasianus*), the "rustling" of wings differs between individuals (*Koch, Krakauer & Patricelli, 2015*). Therefore, mechanical sounds can also carry acoustic information about individuality.

Vocal individuality in some non-passerine groups has been intensively studied, such as colonial birds or nocturnal birds. On the other hand, gallinaceous species has not been

**Table 2 Descriptive statistics and Potential for individual coding.**

| Variable | DFA | Mean | Min | Max | SE | Kruskal–Wallis | Mean CVw | CVa | PIC |
|---|---|---|---|---|---|---|---|---|---|
| 1st Quartile frequency | X | 1,444.3 | 0.9 | 3,000.0 | 271.8 | * | 38,503.6 | 73,887 | 1.919 |
| Relative 1st quartile frequency | X | 3.3 | 0.0 | 1500.0 | 62.4 | * | 219.9 | 3,896 | 17.714 |
| 3rd Quartile frequency | X | 197.8 | 0.1 | 3,027.8 | 436.6 | * | 70,423.4 | 190,607 | 2.707 |
| Relative 3rd quartile frequency | | 0.3 | 0.0 | 1.0 | 0.2 | * | 0.0 | 0 | 1.960 |
| 1st Quartile time | | 1,747.2 | 0.5 | 3,562.5 | 367.6 | * | 60,556.2 | 135,130 | 2.231 |
| Relative 1st quartile time | | 2.4 | 0.1 | 1,687.5 | 56.2 | * | 229.5 | 3,159 | 13.761 |
| Relative 3rd quartile time | | 246.2 | 0.2 | 4,3525.0 | 1508.0 | * | 216,894.9 | 2,274,186 | 10.485 |
| Aggregate entropy | X | 3.2 | 0.7 | 4.8 | 0.5 | * | 0.2 | 0 | 1.846 |
| Average entropy | X | 2.8 | 1.7 | 4.0 | 0.4 | * | 0.1 | 0 | 1.869 |
| Bandwidth 90% | | 839.2 | 187.5 | 2,437.5 | 355.2 | * | 69,079.2 | 126,142 | 1.826 |
| Center frequency | | 1,589.0 | 468.8 | 3,375.0 | 303.2 | * | 45,489.3 | 91,910 | 2.020 |
| Center time | | 198.1 | 0.1 | 3028.1 | 436.6 | * | 70,428.0 | 190,611 | 2.706 |
| Relative center time | X | 0.5 | 0.1 | 1.0 | 0.2 | * | 0.0 | 0 | 2.470 |
| Call duration | X | 1.0 | 0.2 | 1.9 | 0.2 | * | 0.0 | 0 | 1.460 |
| Duration 90% | X | 0.7 | 0.1 | 1.6 | 0.2 | * | 0.0 | 0 | 1.657 |
| Frequency 5% | X | 1,209.8 | 375.0 | 2,250.0 | 249.9 | * | 42,818.7 | 62,427 | 1.458 |
| Relative Frequency 5% | | 0.2 | 0.0 | 0.5 | 0.1 | * | 0.0 | 0 | 1.303 |
| Frequency 95% | | 2,049.0 | 1,125.0 | 4,125.0 | 432.2 | * | 84,005.1 | 186,777 | 2.223 |
| Relative frequency 95% | | 0.7 | 0.3 | 1.0 | 0.1 | * | 0.0 | 0 | 1.823 |
| Inter-quartile range | X | 301.8 | 86.1 | 1,218.8 | 188.9 | * | 17,712.6 | 35,694 | 2.015 |
| IQR (Inter-quartile range) duration | X | 0.4 | 0.0 | 1.2 | 0.1 | * | 0.0 | 0 | 2.301 |
| Max entropy | | 3.9 | 2.9 | 4.9 | 0.3 | * | 0.0 | 0 | 2.044 |
| Max frequency | | 1,572.4 | 468.8 | 3,468.8 | 336.3 | * | 58,737.8 | 113,076 | 1.925 |
| Max time | | 197.9 | 0.1 | 3,027.5 | 436.6 | * | 70,409.4 | 190,584 | 2.707 |
| Min entropy | | 1.6 | 0.1 | 2.9 | 0.4 | * | 0.1 | 0 | 1.685 |
| Peak time | | 197.9 | 0.1 | 3,027.5 | 436.6 | * | 70,409.3 | 190,583 | 2.707 |
| Time 5% | X | 197.9 | 0.0 | 3,027.5 | 436.5 | * | 70,738.2 | 190,565 | 2.694 |
| Relative time 5% | | 0.1 | 0.0 | 0.4 | 0.0 | * | 0.0 | 0 | 2.247 |
| Relative time 95% | | 0.8 | 0.6 | 1.0 | 0.1 | * | 0.0 | 0 | 2.143 |

**Note:**
(DFA) 13 variables included in final DFA model (X). (SE) standard error of the mean. (Kruskal–Wallis) Kruskal–Wallis test after Bonferroni correction, (*) $p < 0.001$. (Mean CVw) within individual comparison. (CVa) between individual comparison. (PIC) Potential for Individual Coding.

studied frequently. Acoustic displays of the Japanese quail (*Coturnix coturnix japonica*) are characterized by a potential for vocal individuality in terms of temporal parameters. Spectral characteristics of the voice are then associated with the possibility of greater stability during the development of the individual, which is important in the question of long-term recognition of individuals (*Sezer & Tekelioglu, 2010*). Call analyses of European and Japanese quail (*Coturnix c. japonica, C. c. coturnix*) confirm a difference between these two subspecies based on the time structure of vocalization (*Collins & Goldsmith, 1998*). The hazel grouse (*Bonasa bonasia*), studied in Switzerland, exhibits 6 to

11 elements of singing during flight. These elements (individual tones or syllables) are characterized by their individual specificity (*Mulhauser & Zimmermann, 2003*). Specific parameters responsible for acoustic individuality were also found in males and females of the common quail (*Coturnix coturnix*); the results of this study also indicated that the male's inter-individuality is dependent on sexual maturation and age (*Guyomarc'h, Aupiais & Guyomarc'h, 1998*). Our study demonstrates a vocal individuality in gallinaceous species with lek mating system.

What role acoustic individuality plays in the black grouse's voice is still a question for future research. Calls of individual birds may carry information about male quality (*e.g.*, physiological state, age) for females (*Guyomarc'h, Aupiais & Guyomarc'h, 1998*), and, at the same time, it might be a signal for other males providing information about the strength of a rival. Finally, individuality can serve to easily identify individuals among each other within a group. Its potential for scientists lies in the possibility of use for noninvasive monitoring. Taking an observation, census may be inaccurate; due to overflights of individuals within the lekking site, repeated census of the same individuals may occur and therefore the results of counting may be overestimated. Monitoring based on acoustic recognition could provide the required accuracy and assistance in areas where observation is limited by environmental conditions (*e.g.*, the situation when males of black grouse lek individually hidden in the undergrowth).

## CONCLUSION

The black grouse population is affected by many factors that contribute to its decline, and as part of its conservation, efforts are being made to develop better methods of protection, including monitoring. Vocalization recording and analysis could be a non-invasive monitoring tool, especially if there is individuality in the voice of individuals. This method could significantly reduce the risk of multiple counting of the same individual. Surprisingly, we found this individuality in the black grouse in the non-vocal type of display. The discrimination model classified each call with high accuracy and important variables turned out to be Frequency 95% and Quartile 1 Frequency. In comparison to the signals produced by the syrinx, non-vocal sounds have been studied rarely and according to our knowledge, this is the second evidence of vocal individuality in avian hissing sounds which are not produced by syrinx. Finding specific identifiers in vocalization could lead to a more accurate determination of the number of individuals.

## ACKNOWLEDGEMENTS

We very much appreciate the help and support of Mr. Christian Krogell, Department of Natural Resources, Ministry of Agriculture and Forestry, Finland. We are also thankful to Mr. Ilkka Ala-Ajos for logistical support for our research in Finland. Furthermore, we would like to thank Dr. David Parish, Marlies Nicolai and Dr. Kathy Fletcher from the Game & Wildlife Conservation Trust and the gamekeepers in Cairngorm National Park, Scotland. We would like to express our gratitude to Robert B. Davis for correction of the English language. Thanks for help in translating articles also goes to Mr. Roland Kralik.

### Funding

This research was funded by the IGA of the Faculty of Forestry and Wood Sciences, Czech University of Life Sciences, Prague. Project number: B_04_18. There was no additional external funding recieved for this study. The funders had no role in study design, data collection and analysis, decision to publish, or preparation of the manuscript.

### Grant Disclosures

The following grant information was disclosed by the authors:
IGA of the Faculty of Forestry and Wood Sciences, Czech University of Life Sciences, Prague: B_04_18.

### Competing Interests

The authors declare that they have no competing interests.

### Author Contributions

- Lucie Hambálková performed the experiments, analyzed the data, prepared figures and/or tables, and approved the final draft.
- Richard Policht conceived and designed the experiments, performed the experiments, analyzed the data, authored or reviewed drafts of the paper, and approved the final draft.
- Jiří Horák performed the experiments and approved the final draft.
- Vlastimil Hart conceived and designed the experiments, performed the experiments, authored or reviewed drafts of the paper, and approved the final draft.

### Animal Ethics

The following information was supplied relating to ethical approvals (*i.e.*, approving body and any reference numbers):

The Animal Care and Use Committee of the Czech Ministry of the Environment approved the protocol (Permit number: 15106/ENV/14-825/630/14).

### Field Study Permissions

The following information was supplied relating to field study approvals (*i.e.*, approving body and any reference numbers):

Field experiments were approved by the Department of Natural Resources, Ministry of Agriculture and Forestry, Finland and by the Game & Wildlife Conservation Trust, Scotland.

### Data Availability

The raw data are available as Supplemental Files.

The resulting model of statistical analysis shows all individuals and significant variables tested. The output of classification results shows classification accuracy percentages for each individual.

## Supplemental Information

Supplemental information for this article can be found online at http://dx.doi.org/10.7717/peerj.11837#supplemental-information.

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
