# Peer review of "Acoustic individuality in the hissing calls of the male black grouse (Lyrurus tetrix)"

_PeerJ, doi:10.7717/peerj.11837_

## Round 0.1 · original submission · Major Revisions

While both reviewers felt that the manuscript was interesting and the writing was generally strong, there are a number of concerns that need to be addressed. Please see especially the comment of both reviewers regarding the need to include a table listing and providing more information for the 29 variables measured in the study, and the comments of reviewer 1 regarding both the ratio of variables to cases and the possibility of running a binomial test to see if the performance of the classification is above that expected by chance. Please also address the concerns of reviewer 1 regarding the experimental design.

Reviewer 1 ·

Basic reporting

The manuscript is concise and well written overall, and addresses an interesting question, that is, whether calls of a bird of the Galliformes order are individual specific.

There are however a number of points that should be clarified and expanded on, especially in the method and result section.
It would be helpful to know which variables were measured (could be included as Table). How where they selected? This should be explained in more detail. In addition, 29 variables are quite a lot. Should not the number of variables be smaller than the number of cases (here 24)? Similarly, the result section is very short, and the abbreviations are not explained (e.g., “AggEnt”, “Dur90%” etc) . Is the performance of the classification really above chance? This could be tested with a binomial test.

The figure captions should be expanded. Figures 1-4 do not only show spectrograms but also oscillograms etc. In Figure 5, please explain the abbreviations. Are figures 1 and 2 really needed? Maybe one figure could depict two recordings from Scotland and two form Finland. Similarly, Fig 3 and 4 could be combined. It would also be helpful if Figure 5 was using different colours to represent calls coming from one individual.

The abstract is not really fitting to the rest of the manuscript: the text is framed around the question of conservation, which I think is really interesting. The abstract however suggests that the paper can speak to the question whether black grouses can recognise individuals, or what they can infer from that information (e.g., lines 34f - this sentence would be more suitable in the discussion). It would be better to stay closer to the text in the abstract.

One important point the authors make is that hissing calls are non-vocal sounds. For someone not knowing much about black grouse, this is confusing. it would be helpful to clarify / explain this aspect in more detail.

Lastly, I would suggest moving the first paragraph of the discussion to the introduction. A discussion should start with a short recap of the aim of the study, and then summarise the findings. At the moment, it starts with a discussion of the different call types, which would be more appropriate in the introduction. The motivation for studying individuality in this species is nicely laid out, however it would be helpful to specify how much the population has decreased lately (line 46).

Experimental design

My major concern is with the data collection and how this might have influenced results: the authors recorded hissing calls from males from different display sites to ensuring that they were indeed recording from different individuals. However, although this procedure makes sense to establish the relevant call parameters, it is not clear whether you are measuring individual or location differences. Thus, it is possible that future studies using these parameters cannot differentiate between individuals from the same location. It would thus be necessary to classify individuals from the same location to establish true individuality.

Validity of the findings

In order to really say that the black grouse can be identified by its hissing call, the authors should test whether there was less variation within calls of the same individual than between individuals. In addition, as I have mentioned above, it is not clear whether the findings represent individual or lek differences.

Additional comments

Some minor comments:
- Throughout the text: should black grouse really be capitalised?
- Line 44f: consider re-writing as “Although the black grouse is classified as a species of least concern..”
- Line 47: replace “reasons” with “causes”
- Line 64: replace “so” with “thus also”
- Line 87f / 106f: these two sections mentioning ethics and licenses could be combined
- Line 96: Please explain here what a lek is
- Line 132: remove the “n” in ”(~n < 1)”
- Line 143: this information is already provided in the method section
- Line 144: It would be helpful to have access to the table in the manuscript, as opposed to as supplement
- Line 160: “but we do not suppose” sounds odd

Reviewer 2 ·

Basic reporting

The language was clear, the experiment was put in the context of the literature and figures and data were available. However, there were a number of paragraphs that I was not sure why they were in the manuscript. They were clearly written but seemed to be more examples trying to support a point, though I was often unsure of what point was being supported or why that point was being highlighted in the first place.

Experimental design

Question was clear, why this study was important was clear, and the methods were solid. However, I would have liked to see all of the variables tested and which ones were correlated.

Validity of the findings

The conclusions follow from the findings. The discussion was a bit meandering for me, and I was not sure why some paragraphs were included, but the writing was clear and didn’t overstate the results.

Additional comments

42-63: I think that this paragraph could be shortened and streamlined a bit, making more general statements about sources of mortality rather than outlining many specific examples.

73-76: here also, it could be shortened, just focusing on the fact that this has shown a lot of promise in other studies but not explaining in detail why and how. Additionally, by including the extra information about the yearling’s song, it undermines the argument that vocal monitoring is a good idea (try focusing on the potential useful possibilities, not on past failures).

77: ‘repose’ is a slightly odd choice of word here, I am not quite sure what is meant by tis sentence. Maybe try thinking of another one.

120-121: what variables? I would like to see these maybe in a table with an explanation for what each variable is, and which ones were highly correlated.

147-152: how were the males that were difficult to classify with accuracy specially patterned? Sometimes theories of individual vocal identification suggest that within close groups/neighbours there is a lot of individual differences, but that this may change as samples from farther away are taken.

155-156: I am a bit confused by this sentence as it starts out with 4 call types, then after 2 suggests that there are 4 more call types?

187-200: while this paragraph has many examples of species in regard to differences in vocalizations, I am not sure what the point of this paragraph is. Please make it a bit clearer in the first few sentences and a last sentence.

213-220: unless this is a specific section included in all PeerJ articles, I think ending without it is stronger.

---

## Round 0.2 · Minor Revisions

On the whole, this revision is very strong. There are really just two points that still need some clarification (see points 1 and 6 below) and a few minor suggestions to strengthen the writing and cut repetition.

1. Most importantly, please respond thoroughly to reviewer 1’s concern about the experimental design.

2. Line 49 (last sentence of abstract) – this phrasing is awkward. Please change it to something like “where some individuals are overcounted as multiple individuals”. If this is not what you meant then please rephrase what you currently have to be clearer.

3. Line 54 – consider changing “its actual number of individuals” to “determining the actual number of individuals”

4. Line 82 – “this decline” would generally refer to something you mentioned in the previous sentence, but that sentence doesn’t specifically refer to a decline (it says “decrease not exceeding 30%”, which could be 0%). If the references in the first sentence refer to a specific decline then I would suggest putting that sentence directly before the “this decline” reference instead.

5. Last paragraph of section on ”study areas and recording”. You have redundant information in here about how many individuals were selected per site: “We selected only one individual from each display site to record. In several localities, we recorded two individuals, but we always selected males from opposite sides of the display sites.” And “To avoid the risk of multiple counting of the same individual, we chose the option of selecting only one, maximum of two individuals on each lek.”

6. Second sentence under “Statistical analyses” Was this 7 out of the 31? Or 7 out of an original 38, leaving 31? Please clarify.

7. Fourth sentence under “Statistical analyses” – you already had virtually the same sentence “We excluded variables with low or no variance.” at the end of the previous section. Please remove one of these.

Reviewer 1 ·

Basic reporting

-

Experimental design

My only remaining question refers to the ratio of variables to cases - the authors response does not really addresses my problem, as there are still only 24 cases? Or when did the exclusion happen, after the analysis? To me that is not clear. In addition, it would be good to know more on how / why individuals were excluded - e.g., what does "extreme" mean?

Validity of the findings

-

Additional comments

I am overall happy with the revisions made except see above.
Minor issues I noticed:
- L 151: include that CV means “coefficient of variation” (maybe like this: “..parameter, we compared the coefficient of variation (CV) within and between individuals.”)
- L 173: is there a “the” missing?
- I am not sure I understand table 2. Is the X in the second column indicating significant variables?

Reviewer 2 ·

Basic reporting

The language was clear (barring a few grammatical issues), the experiment was put in the context of the literature and figures and data were available. The intro and discussion were more to the point and straightforward.

Experimental design

Question was clear, why this study was important was clear, and the methods were solid. I was happy to see the inclusion of the table of the variables tested.

Validity of the findings

The conclusions follow from the findings. The discussion was more straightforward, and the writing remained clear and didn’t overstate the results.

Additional comments

47-48: this sentence makes it sound like this decline is in the past. Maybe something more like “as black grouse continue to decline across its distribution range, . . . “

49-52: this sentence makes it sound like the decline isn’t too worrying, so why then bring it up? Maybe leave out the ‘classified as a species of least concern’ and focus on the decline, or ‘though it is classified as a species of least concern . . . it continues to decline across its range requiring monitoring.’

124-126: how many individuals were there on a lek? Why not look at individuals within leks to make sure they were different, rather than between leks since between leks you can tell different males are present due to site fidelity?

240-251: this sounded a bit like a very short abstract rather than a conclusion. The last 3 sentences were more like a conclusion.

---

## Round 0.3 · accepted · Accept

Thank you very much for your latest revisions, which both I and the reviewers feel improved the paper further.